# Appropriability and basicness of R&D: Identifying and characterising product and process inventions in patent data

**Sebastian Heinrich**[1☯¤], **Florian Seliger**[1,2]*, **Martin Wörter**[1☯¤]

**1** KOF Swiss Economic Institute, Zurich, Switzerland, **2** NZZ Media Group, Zurich, Switzerland

☯ These authors contributed equally to this work.
¤ Current address: KOF Swiss Economic Institute, ETH Zurich, Zurich, Switzerland
* seliger@kof.ethz.ch

## Abstract

We present a database that classifies all patent applications filed at either the United States Patent and Trademark Office (USPTO) or the European Patent Office (EPO) as being either product patents, process patents or 'mixed patents'. We use the share of claims that refer to either product or process inventions which allows to classify all patent applications along a continuum of pure process patents and pure product patents. We find that process-oriented patents draw more on previous knowledge, are more original and more radical than product patents. Lower breadth of protection is positively associated with pure process patenting, whereas product and mixed variants can be protected more broadly. This characterisation uncovers heterogeneity of patented inventions that allows for a more sophisticated use of patent statistics. It can improve the accuracy of analyses, but also reveal new aspects related to property rights.

## Introduction

In this study, we present a categorization of patents according to whether they reflect product or process inventions. We identify keywords in patent claims that are related to process inventions, which allows us to classify the universe of European Patent Office (EPO) and United States Patent and Trademark Office (USPTO) patents. We also characterize process and product patents based on fundamental patent characteristics. In particular, we identify differences in product and process inventions by simultaneously regressing them on indicators capturing appropriability and basicness of R&D.

The distinction between product and process patents and their characterization contribute to several empirical and theoretical research strands in the field of economics.

For instance, the cost-saving function of process inventions is a cornerstone in industrial economics [1], especially when analyzing market power [2], market entry [3], and innovation behavior as a function of competitive conditions [3–10], or when analyzing the impact of cost-saving inventions on the labor market [11]. The distinction between process and product inventions in patent statistics thus allows to test more theoretical models empirically.

**Data Availability Statement:** The data are available to the public on Harvard Dataverse (DOI: https://doi.org/10.7910/DVN/CBSK2W).

**Funding:** We are grateful to the European Patent Office Academic Research Programme for the

financial support of this project. Part of Sebastian Heinrich's salary was funded by this grant. The funders had no role in study design, data collection and analysis, decision to publish, or preparation of the manuscript.

**Competing interests:** The authors have declared that no competing interests exist.

The differentiation between product and process technologies also plays a key role for technological life cycles. Utterback and Abernathy (1975) [12] described the stylized properties of such life cycles, where product inventions dominate at the beginning when firms compete with a variety of products. Firms first try to be competitive by introducing new products at an early stage of a life cycle. As technologies mature, they compete mainly on prices, while products are changed only incrementally. At this stage, competitive pressure induces firms to reduce production costs. In order to remain competitive and to sell into mass markets, firms then introduce process inventions. In the same vein, Klepper (1996) [13] showed that the ability to appropriate returns from process R&D depends on firm size. When firms grow in mature industries, they have an increased incentive to leverage process innovation. Finally, Pisano (1997) [10] showed that firms' investment in new products are followed by investment in processes in order to decrease production costs in price competition.

Regarding patenting behavior, firms are assumed to follow different strategies for product and process technologies. According to Cohen and Klepper (1996) [13], firms typically do not sell or license process patents within their industry. One reason might be that process patents are less effective in protecting the underlying intellectual property. In general, the leaking of knowledge to competitors is slower as with product patents which reduces the incentive of patenting [14]. Hall et al. 2010 [15] provide a review on what determines firms' choice between patenting and secrecy to protect their intellectual property. They conclude that innovations' characteristics, namely product versus process, as well as discrete versus complex, along with competition in the product market, are key factors for the propensity of choosing patenting or secrecy.

Many empirical studies use the framework of a knowledge production function to investigate the returns from knowledge accumulation within a company [16–19]. They often use patent data to measure the knowledge stock, lacking the distinction between product and process-related knowledge. Hence, it is an open question whether potentially positive returns from knowledge accumulation come from process or product-related knowledge. So far, there have been smaller scale empirical studies that distinguish between those two types of knowledge. They focus on specific technologies, on specific types of companies [20, 21] or use survey information [22].

Despite the important role of the distinction between product and process related knowledge in economic theory, technological life cycles, the knowledge production function, and for patent strategies, there are relatively few empirical studies investigating process inventions [10, 23–26].

The negligence of process inventions in empirical studies mainly results from the lack of data. One of the few research efforts include Scherer (1982) [27], constructing a patent-based matrix for technology flows between industries. Cohen and Klepper (1996) [13] used the same data to study the distinction between product and process data. Furthermore, there are investigations based on survey data, studying the direct performance effects of product vs. process innovations [22, 28]. There are only a few recent studies that try to distinguish between products and processes in patent claims [29–31].

The main contribution of this paper is to provide a comprehensive categorization of patents according to whether they represent product or process inventions. The patent classification includes patent documents from two major patent offices, the EPO and the USPTO. We make the data available to the public under the following DOI https://doi.org/10.7910/DVN/CBSK2W so that future research can draw on this comprehensive database. The proposed classification allows for more extensive studies analyzing the impact of product and process knowledge on firm performance and thus addresses small sample issues [19].

Our descriptive analysis shows that product patents are much more common than process patents, but 'mixed' patents (patents with both product and process claims) have become more important over time. In many technologies, mixed patents have become the predominant form of patenting already in the nineties. We see rather large differences across countries and technologies. This reveals previously undiscovered differences in countries' technological capabilities and orientations, which can increase our understanding of the importance of technologies for countries' economic development.

A further contribution refers to the investigation of the relationship between fundamental patent characteristics and the likelihood that a patent is a pure process patent or mixed patent rather than a product patent. Given that there is only a vague understanding of how process patents differ from product patents and its importance for economic thinking, it is crucial to gain more knowledge about the differences from an empirical point of view.

The characteristics we investigate are indicators that are commonly used in patent studies and are thus well-established. We mainly focus on appropriability and basicness of R&D. According to Trajtenberg et al. (1997) [32] they are prominent sources of heterogeneity in the characterisation of R&D. We therefore expect that they are also essential for the disctinction between product and process inventions. Basicness refers to features of innovations such as originality or closeness to science, whereas appropriability refers to the ability of inventors to reap the benefits from their inventions.

Our empirical study shows that patents with a higher share of process claims seem to be in younger technologies and have a broader technological scope compared to product patents. They draw on previous knowledge to a larger extent than product patents and are more radical and original. Finally, smaller breadth is associated positively with process patenting, but negatively with patenting mixed types, which is partly in contrast to our expectations.

The first section explains the applied method to distinguish between product and process patents. The second section presents descriptive results and shows how the number of product and process patents has evolved over time. In the third section, we describe the variables we use to identify important differences between patent types and present econometric results of a multinomial logit estimation. The last section presents conclusions from this study.

## Identification of product and process patents

Patents have multiple claims (in most of the cases between 5 and 50), and we classified each claim individually as belonging to either the process or the product class. Claim texts for EPO patents were obtained from the EPO backfile containing EP-A and EP-B documents from 1978 to 2016 and 1980 to 2016, respectively (see https://www.epo.org/searching-for-patents/data/bulk-data-sets/data.html#tab-2, accessed on 2019/09/27). Claim texts for USPTO patents were obtained from the 'Patent and Patent Application Claims Research Dataset' provided on the USPTO's Bulk Data Storage System containing full-text claims from US patents granted between 1976 and 2014 and US patent applications published between 2001 and 2014 (see https://bulkdata.uspto.gov/, accessed on 2019/09/27). We focus on USPTO and EPO patent documents as they belong to the five largest patent offices in the world, therefore covering a great share of global patent families. The EPO was founded in 1977, so data has only become available in the aftermath. The end year was also chosen due to data availability at the time we began to work on the project. The introduced classification scheme has been developed within the European Patent Office Academic Resarch Programme [33].

The set of keywords we use to classify claims were determined by a manual search process in EPO patents. The search was done by a patent expert from the Fraunhofer Institute in Germany. However, the identification of product patents by keywords is incomplete as in many

cases the labeling of the specific product is used instead of more abstract terms. In contrast, the identification of processes is quite complete if using the extracted keywords. Since patent applications at the EPO can be filed in English, German or French, and not all patent filings are available in English, we had to conduct keyword searches in all three languages because we apply the keyword search on all EPO documents.

In contrast to economic theory, where a process invention is usually an improved process to produce a good (which mostly implies cost reductions), the definition of process claims is much broader. The EPO examination guidelines include "all kind of activities in which the use of some material product for effecting the process is implied" [34]. According to the USPTO guidelines, process claims "include a new use of a known process, machine, manufacture, composition or material" [35]. In order to account for those "use" claims, we identified additional keywords that refer to the usage of something. In this paper, our definition of process claims comprise both processes, methods and use claims. Our database enables researchers to distinguish between processes and usages explicitly, but the share of use claims is too tiny in order to be relevant for statistical analyses. The extracted keywords to distinguish between product and process claims for all three languages are listed in Table 1. The latter includes typical process plus use claims. For USPTO patents, we use the same set of keywords only in English.

We implemented the classification as an exclusion process, meaning that every claim was checked for the occurrence of one of the process keywords. If one of the keywords occurs, the claim is considered as belonging to the process class. On the contrary, if a claim does not contain any of the process keywords, it is considered as belonging to the product class. This keyword search takes into account that the set of product keywords is incomplete and therefore does not use them explicitly. In this way, an unambiguous classification of claims is possible: If a claim is not a process claim, it must be a product claim by definition (see also the examination guidelines [34, 35]). So-called 'product-by-process' claims are product claims as well, but the difference to a standard product claim is that the product they relate to is defined by the process for producing it. The classification of claims offers a granular classification of each patent by considering that a substantial part of patents contains both products and processes.

In order to apply the keyword search, we had to apply some heuristic rules. Most importantly, we had to limit the search to the first words of a claim. Otherwise we would have classified many product claims as processes by mistake. Details and examples are provided in (S1 File). For process claims, we use two different sets of rules. In the first one, we limit the keyword search to the first two words. In the second, we limit the search to the first five. In turn, for use claims, we limit the search to the first word of a claim.

After the classification had been completed, we computed the total number of claims, the number of product, process, and use claims, as well as the share of process claims and use claims for each patent. These numbers serve as the basis for the identification of product and process patents.

In order to ensure the validity of the applied classification, we mainly did two things: First, we classified thousands of claims manually and compared the classification result with results from the automated keyword search. Second, we repeated the classification using machine learning. In order to apply machine learning techniques, research assistants who studied engineering or natural sciences classified the claims from about 1'100 patents that have been granted and were randomly drawn from the Worldwide Statistical Patent Database (PATSTAT). This served as our training data. For details, see Banholzer et al. (2019) [33]. We compared the accuracies from keyword search and text mining where the keyword approach yielded a slightly higher accuracy (98 compared to 93). By extensive manual inspection, we also came to the conclusion that the keyword search approach was superior in capturing the

**Table 1. Keywords for classification into product and process claims.**

| | Typical product keywords | Process keywords | Use keywords |
|---|---|---|---|
| *English* | device | method | use |
| | machine | process | utilization |
| | material | procedure | utilisation |
| | tool | | usage |
| | apparatus | | |
| | vehicle | | |
| | compound | | |
| | composition | | |
| | substance | | |
| | article | | |
| *German* | Vorrichtung | Verfahren | Verwendung |
| | Einrichtung | Methode | Anwendung |
| | Werkzeug | Prozess | Benutzung |
| | Material | Prozedur | Nutzung |
| | Apparat | | |
| | Fahrzeug | | |
| | Verbindung | | |
| | Zusammensetzung | | |
| | Substanz | | |
| | Artikel | | |
| *French* | outil | procédé | utilisation |
| | machine | méthode | usage |
| | support | procedure | |
| | materiel | processus | |
| | dispositiv | | |
| | assemblage | | |
| | véhicule | | |
| | composé | | |
| | composition | | |
| | substance | | |
| | article | | |

distinction between the different classes. However, the two measures appear to be correlated to a very large extent when comparing their outcomes.

## Results from keyword search

The use share per patent is very small and does not change significantly over time. Therefore, we will add the use share to the process share in this paper. This is in line with the examination guidelines that treat claims referring to the use of something as process claims. As already mentioned, product-by-process claims are in fact product claims. Consequently, we treat them as product claims throughout the analysis, i.e. we add them to a patent's product share.

Concerning the process share, there has been a trend towards including more process claims since 1990. The share amounts to about 30% nowadays (about 22% in 1980). There is only a slight difference between process shares based on keyword searches in the first two words and in the first five words. In this paper, we use the more restrictive definition based on only the first two words.

## Definition of product and process patent categories

For the subsequent analyses, we use a rather restrictive definition of process and product patents: If a patent filing only contains product claims (i.e., the share of product claims is 1), it is considered a product patent. If it only contains process or use claims or process and use claims at the same time, it is defined as process patent. Patents with both product and process-use claims are defined as 'mixed patents'. This definition is admittedly narrow and we lose information on the exact share of process and product claims in mixed patents because there is a continuum of possibilities between the two polar cases 'pure' process patents and 'pure' product patents. However, it has the advantage of being clear-cut and that we do not need to establish arbitrary thresholds (such as a 50% rule etc.).

## Descriptive results

In this section, we provide descriptive results for the development of product, process-use and mixed patents based on patents granted at either the EPO or USPTO with priority years between 1980 and 2010.

Figs 1 and 2 show the number of granted product, process, and mixed patents at the EPO and USPTO, respectively. At the EPO, the share of pure product patents is around 50%, at the USPTO, it is lower. The share of pure process patents is generally much lower and slightly decreasing at both offices. In contrast, the number of mixed patents has increased considerably over time. It has caught up with the number of product patents in recent years.

There are hardly any studies that we can refer to in order to validate our results. An exception to this is an EPO study on the 'Market success for inventions', which presents results from interviews with SMEs on the type of a specific patented invention [36]. The shares of pure product, pure process, and mixed patents are very close to our figures (according to the survey, 47% of the patent applications refer to pure product inventions, 38% to inventions combining product and process features, and 15% to pure process inventions). The survey results might increase confidence in the classification approach and help alleviate concerns about the use of claim text (e.g., because they might reflect the examiners' point of view rather than the firms' inventions).

Looking at the average number of claims per patent, we can see that the number has increased from about 9 to 12 at the EPO and from 11 to 17 at the USPTO from 1980 to 2010. Claims can be distinguished between independent and dependent claims. The difference is that a dependent claim cannot stand alone, this means it references another claim (independent claim) that is directed to the essential features of the invention. Interestingly, the number of independent claims, which are the claims that can stand alone, has not changed by much; especially at the EPO, it does not show any upward trend. The increase in the number of dependent claims might be driven by strategic reasons, such that firms are trying to make their patents as broad and vague as possible in order to sue competitors that infringe the patent, or by legal requirements at the patent offices. Therefore, indicators based on independent claims might get closer to 'true' product or process shares of inventions. The share of product patents calculated based on only independent claims is higher at the EPO (between 61% and 67%, Fig 3). At the USPTO, product patents have lost their predominant position, but they are still more important than mixed patents when looking at only independent claims (Fig 4).

There are also some interesting differences across inventor countries and technologies (see [33]). For complex technologies such as computer technology, the descriptive evidence delivers a clear picture: Mixed patents have become predominant, whereas product patents loose in significance. Pure process patents only play a minor role. The growing importance of

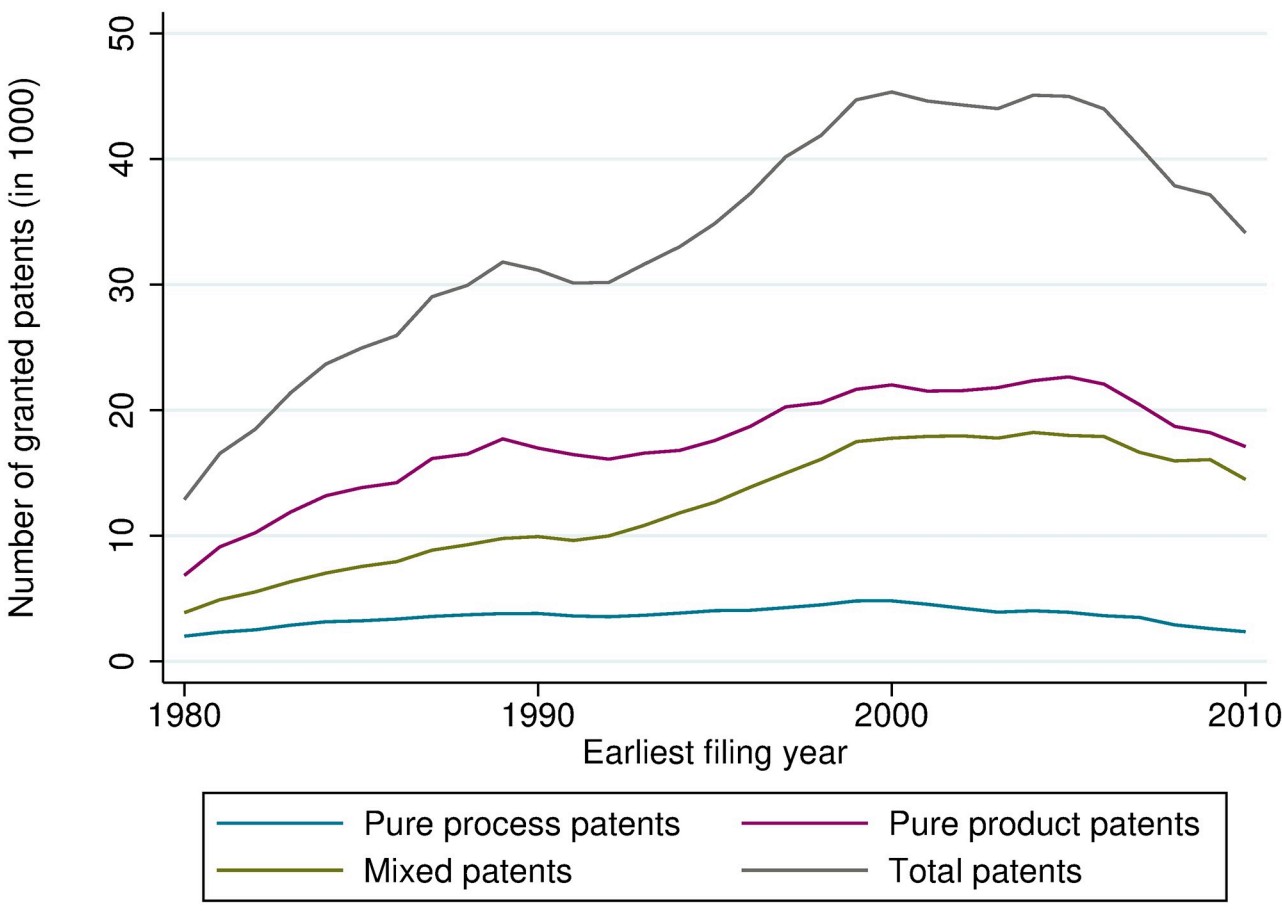

**Fig 1. Count of product, process, and mixed patents, granted at the EPO.**

computer technology in the U.S. might be a reason why we see such a large decrease in product patents at the USPTO.

The reasons for those developments remain unclear and require more detailed analyses, but it is likely that technologies have become more complex over time or that firms add additional process claims in order to fulfil legal requirements or for strategic reasons. The increase in mixed patents might be hardly attributed to cost reductions or improvements in the production process alone.

For technologies that are characterized by very high shares of product patents such as transport technologies, there is also a trend towards more mixed patents (and less product patents), perhaps indicating technological exhaustion within a technological paradigm [37], but the share of product patents is still much higher.

## The relationship between fundamental patent characteristics and product and process patenting

### Data and variables

In this section, we investigate the relationship between fundamental patent characteristics and the likelihood of getting a pure process-use or mixed patent granted in comparison to the likelihood of getting a pure product patent granted at either the EPO or USPTO. This analysis

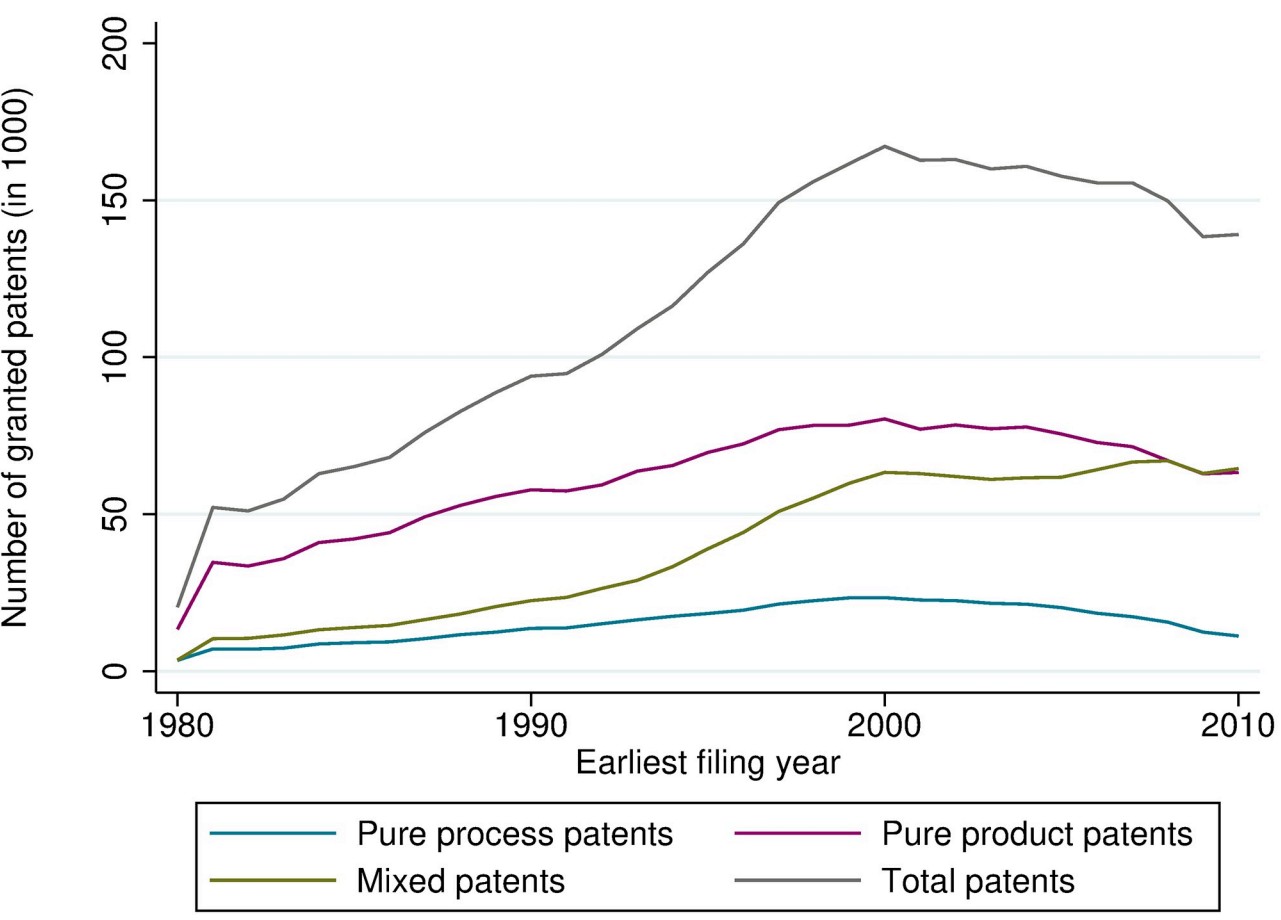

**Fig 2. Count of product, process, and mixed patents, granted at the USPTO.**

enables us to understand how product and process patenting is related to appropriability, basicness, the state of the technological life cycle, and technological scope. The used variables originate from the OECD Patent Quality Database [38], PATSTAT, and the database we have developed as outlined in this study. We provide an overview of our variables and technical description, as well as their sources in Table 2 and provide a summary of their intuition in the following paragraphs.

The following variables, namely patent scope, family size, citations to non-patent literature, backward citations, originality and radicalness come from the quality indicators provided by the OECD [38]). They measure different dimensions of basicness and patent value. *Patent scope* is measured by distinct 4-digit IPC subclasses. A wider scope has been shown to affect valuations of firms where broader patents are more valuable when there are many available substitutes in a particular product class [39]. *Family size* is measured by the number of patent offices where an invention is protected where larger families are associated with higher values [40]. *Non-patent literature citations* can be applied to asses the extent to which a patent relies on scientific research. Patents relying more on scientific research are associated with significantly higher quality [41]. The number of *backward citations* can be used to assess the novelty of an invention disclosed in a patent and was also found to be positively associated with patents' value [40]. *Originality* is measured by the breadth of cited IPC codes and thus measures

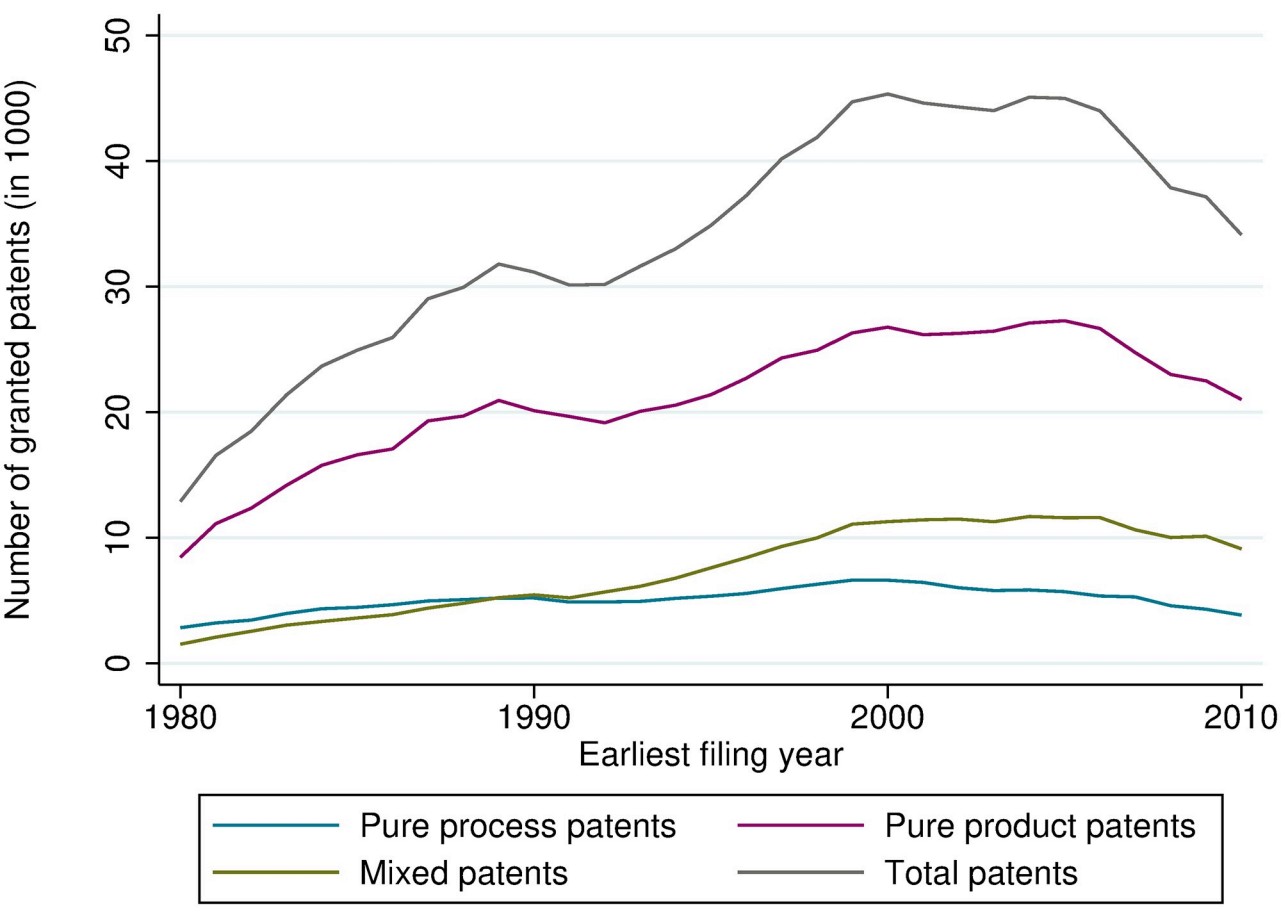

**Fig 3. Count of product, process-use, and mixed patents, EPO, based on independent claims.**

how diversified the knowledge sources are [32]. Finally, the *radicalness* index proposed by Shane (2001) [42] is based on IPC classes that deviate from classes in the cited patents and thus measures the paradigm shift a patent brings about.

Furthermore, we construct a variable intended to capture technological life cycles in the context of product and process patenting. Within a prototypical technological life cycle, a firm devotes more and more effort into process inventions over time [12]. Our *upward technology cycle* variable indicates whether a patent belongs to a growing technology as measured by the growth of the number of patent applications in the respective technology at a given point in time. Another variable measures the *age of a technology*, i.e. the difference between the filing date of the respective patent and the first patent that appeared in a given technology. Technologies are defined as unique combinations of 4-digit codes from the International Patent Classification. A detailed explanation can be found in (S1 File).

Based on our own data, we use the number of *independent claims* to measure a patent's breadth in terms of legal protection. As a higher number of claims incurs higher patent fees [38], they not only account for technical breadth, but also for applicants' expectations of market value [43, 44]. We also add the *number of dependent claims per independent claim* which might reflect strategic considerations of applicants.

A further variable refers to the average *number of words* in the independent claims. According to Kuhn & Thompson (2019) and Marco, Sarnoff, & deGrazia (2019) [45, 46], this is a

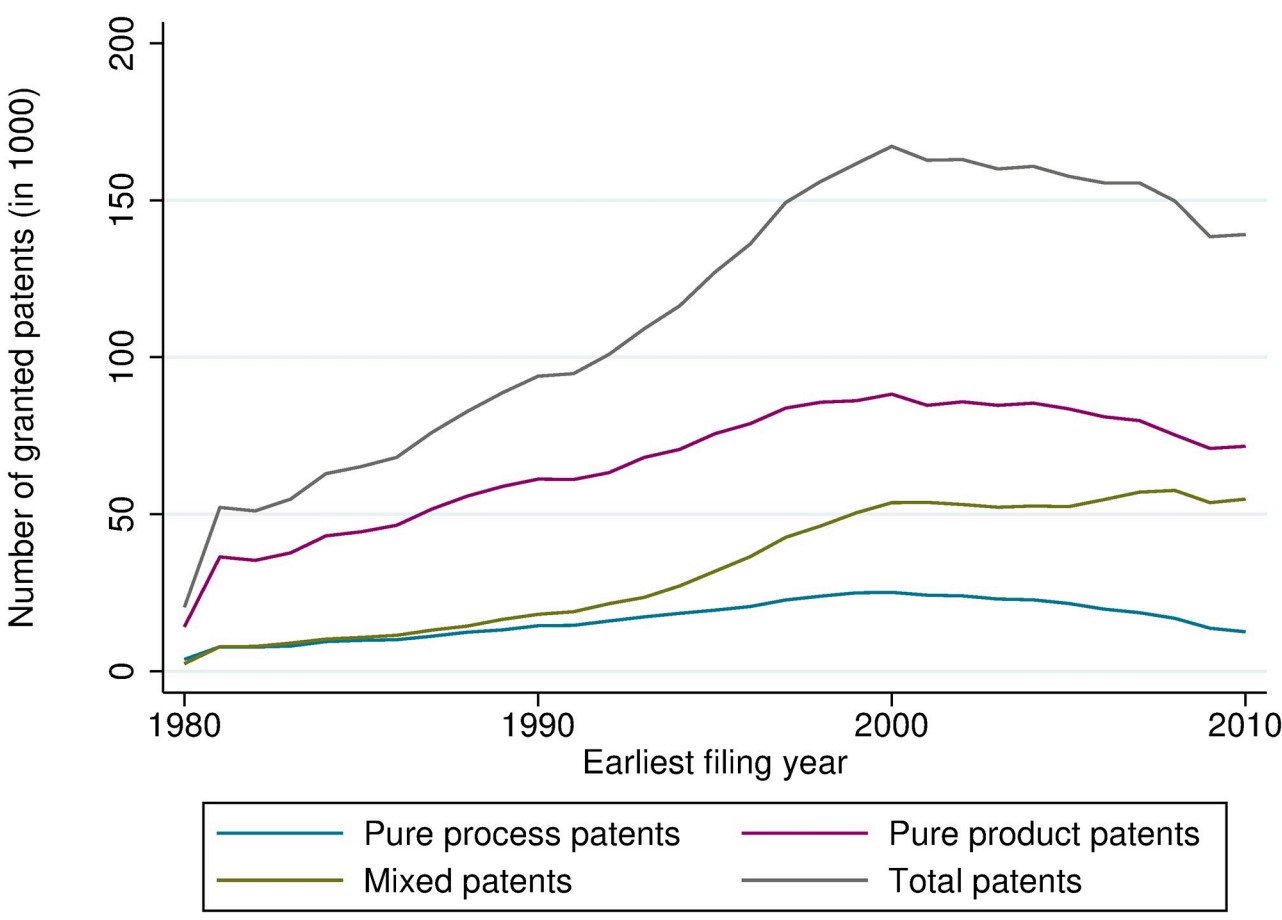

**Fig 4. Count of product, process-use, and mixed patents, USPTO, based on independent claims.**

preferred measure of patent protection breadth. The process of claim narrowing during the examination procedure almost always involves adding words to the claim, i.e., a higher number of words indicates a lower patent breadth. In this paper, we refer to patent breadth in the sense of the breadth of legal protection and to patent scope in the sense of the technological scope of the invention.

Finally, we use the *team size of inventors* to explore potential distinctions in human capital between the two categories. Akcigit et al. (2018) [47] showed that the average team size is 2.3 inventors. Inventor teams are likely to generate more spillovers [48], reduce the cost of accessing shared resources [49], and ultimately increase productivity, especially if policies support the quality of inventor teams [50].

We also add a dummy indicating whether a patent has been filed at the *USPTO* (rather than at the EPO) and the respective inventor country weights and technology field weights. The reason that we add weights rather than dummies is that each patent can belong to multiple inventor countries and technologies. The weights capture the share of inventors in each country and the share of IPC codes belonging to a certain technological field accurately. Finally, we add filing year fixed effects.

Literature on patent quality and appropriability is rather silent with respect to the difference between process and product patents. Most importantly, patent protection of processes is much more difficult than patent protection of products and secrecy is often used as an

**Table 2. List of variables.**

| Variable name | Source | Description |
|---|---|---|
| *product_process_mixed* | Own data | Categorical variable, 1 = product patent, 2 = process patent, 3 = mixed patent |
| *product_process_mixed_ind* | Own data | Categorical variable, 1 = product patent, 2 = process patent, 3 = mixed patent, based on independent claims |
| *lnclaims_ind* | Own data | Number of independent claims, natural logarithm |
| *claims_per_ind_claim* | Own data | Number of dependent claims per independent claim |
| *lnpatent_scope* | OECD Patent Quality | Number of distinct 4-digit IPC subclasses, natural logarithm |
| *lnfamily_size* | OECD Patent Quality | Number of patent offices at which a given invention has been protected normalised with respect to the maximum value exhibited by other patents in the same cohort, with cohorts that are determined by the pair technology–year, natural logarithm |
| *lnnpl_cits* | OECD Patent Quality | Number of NPL citations included in a patent divided by the maximum number of NPL ciations of patents belonging to the same year and technology cohort+1, natural logarithm |
| *lnbwd_cits* | OECD Patent Quality | Number of backward citations included in a patent divided by the maximum number of backward citations of patents belonging to the same year and technology cohort+1, natural logarithm |
| *originality* | OECD Patent Quality | Between 0 and 1, the measure is high if a patent cites patents belonging to a wide range of IPC codes |
| *radicalness* | OECD Patent Quality | Count of IPC-4 digit codes of patent j cited in patent p that is not allocated to patent p, out of n IPC classes in the backward citations counted at the most disaggregated level available, normalised with respect to the total number of IPC classes listed in the backward citations |
| *upward_tech_cycle* | Own data | Patent was filed in a technology when technology was growing (1), when it was stable (0), when it was decreasing (-1) |
| *lntech_age* | Own data | Age of the technology when the patent was filed+1, natural logarithm |
| *lnnumb_words_ind* | Own data | Average number of word stems in independent claims, natural logarithm |
| *lnnb_inventors* | PATSTAT | Number of inventors |
| *USPTO* | Own data | Patent application was filed at the USPTO |

appropriation mechanism for process inventions [15]. Therefore, the breadth of legal protection should be smaller for process and mixed patents patents compared to products patents. On the other hand, we would expect process and mixed patents to be slightly more complex as they dominate in so-called complex technologies. Hence, they should draw on science and predecessor inventions to a larger extent than product patents. They might also cover a larger number of technological fields, might have more claims and a larger number of patent family members. Adding process claims to product inventions and filing mixed patent applications can of course have different implications, e.g. that they are filed for strategic reasons.

Table 3 provides descriptive statistics for the variables used characterize our classification. The dependent variable is a categorical variable (product, process, or mixed patent), containing 51.91% product, 12.25% process and 35.84% mixed patents. When only considering independent claims, the variable contains 57.96% product, 13.95% process and 28.09% mixed patents.

## Methods

Our method to investigate the relationship between the introduced patent characteristics and the patent classification contains two steps. First, we reduce the number of patent characteristics by applying a factor analysis. Second, we use the resulting factors in a multinomial logit model (see [51]) to estimate their relationship with our three patent categories. The following subsection describes the process of extracting the latent factors and their use in the multinomial logit model.

**Factor analysis and multinomial logit model.** With the goal of higher interpretability of the subsequent multinomial logit model, we aim at reducing the number of variables included.

**Table 3. Descriptive statistics of variables.**

| Variable name | Observations | Mean | sd | min | max |
|---|---|---|---|---|---|
| product_process_mixed | 4993150 | 1.839291 | .9228789 | 1 | 3 |
| product_process_mixed_ind | 4992865 | 1.70132 | .8782247 | 1 | 3 |
| lnclaims_ind | 5008417 | .6717842 | .6288613 | 0 | 5.442418 |
| claims_per_ind_claim | 5008417 | 7.423275 | 5.432932 | 1 | 887 |
| lnpatent_scope | 4995389 | .4810146 | .524585 | 0 | 3.688879 |
| lnfamily_size | 4995447 | 1.140025 | .8218295 | 0 | 4.025352 |
| lnbwd_cits | 4995447 | 2.279622 | .9111179 | 0 | 8.467373 |
| lnnpl_cits | 4995447 | .5459317 | .9279244 | 0 | 7.301148 |
| originality | 4890962 | .7088248 | .2202669 | 0 | .9938309 |
| radicalness | 4893208 | .3544737 | .2700929 | 0 | 1 |
| upward_tech_cycle | 4929540 | .737313 | .6754943 | -1 | 1 |
| lntech_age | 4929735 | 2.683784 | .8805059 | 0 | 3.555348 |
| lnnumb_words_ind | 5008415 | 3.657549 | .3697721 | 0 | 7.986845 |
| lnnb_inventors | 5008495 | 1.129451 | .4239406 | 0 | 4.343805 |
| USPTO | 5008495 | .7765179 | .4165788 | 0 | 1 |

We therefore apply a factor analysis (see for example [52]) to find common factors of the patent characteristics introduced above. Our justification to apply a factor model to the patent characteristics relies on the fact that our candidate variables can have a high degree of overlap in their measurement and intuition. For example, a couple of variables rely on claims or citations. This can lead to multicollinearity problems in the estimates, which complicates the interpretation of the results. To avoid this problem, we introduce the orthogonal factors from the factor analysis into the multinomial logit estimation.

We apply the Kaiser criterion [53] and drop all factors with eigenvalues smaller than 1.0 (see the scree plot in Fig 5).

This leaves us with four factors that are presented in Table 4, which shows the rotated factor loadings for the variables *lnclaims_ind*, *claims_per_ind_claim*, *lnpatent_scope*, *lnbwd_cits*, *lnnpl_cits*, *lntech_age*, *originality* and *radicalness*. The absolute loadings bigger than 0.35 are marked in bold, indicating the variables with the strongest loadings on the respective factor.

In the subsequent estimation, the four factors are named as follows: "Citations", with the strongest factor loadings on *lnbwd_cits* and *lnnpl_cits*, "originality and radicalness" with the strongest factor loadings on *originality* and *radicalness*, "age and scope" with the strongest factor loadings on *lnpatent_scope* and *lntech_age*, and "claims", with the strongest loadings on *lnclaims_ind* and *claims_per_ind_claim*. We also estimated additional models, including the full set of our variables and evaluated them against technical criteria (Kaiser criterion) as well as the intuition of the resulting factors.

We apply a multinomial logit estimator to distinguish the importance of the factors for our categorization outcome, specified in Eq (1). In our specification, we use the dependent variable $y \in (0, 1, 2)$ to represent the outcomes product patents ($j = 0$), process patents ($j = 1$) and mixed patents ($j = 2$), where product patents ($j = 2$) are the base category.

$$Prob(Y_i = j \mid x_i) = \frac{exp(\mathbf{x}_i'\boldsymbol{\beta}_j)}{\sum_{j=0}^{2} exp(\mathbf{x}_i'\boldsymbol{\beta}_j)}, \ j = 0, 1, 2 \tag{1}$$

We use the estimations from the above factor model as variables in our multinomial logit estimation with robust standard errors, in the sense of a principle component regression (see

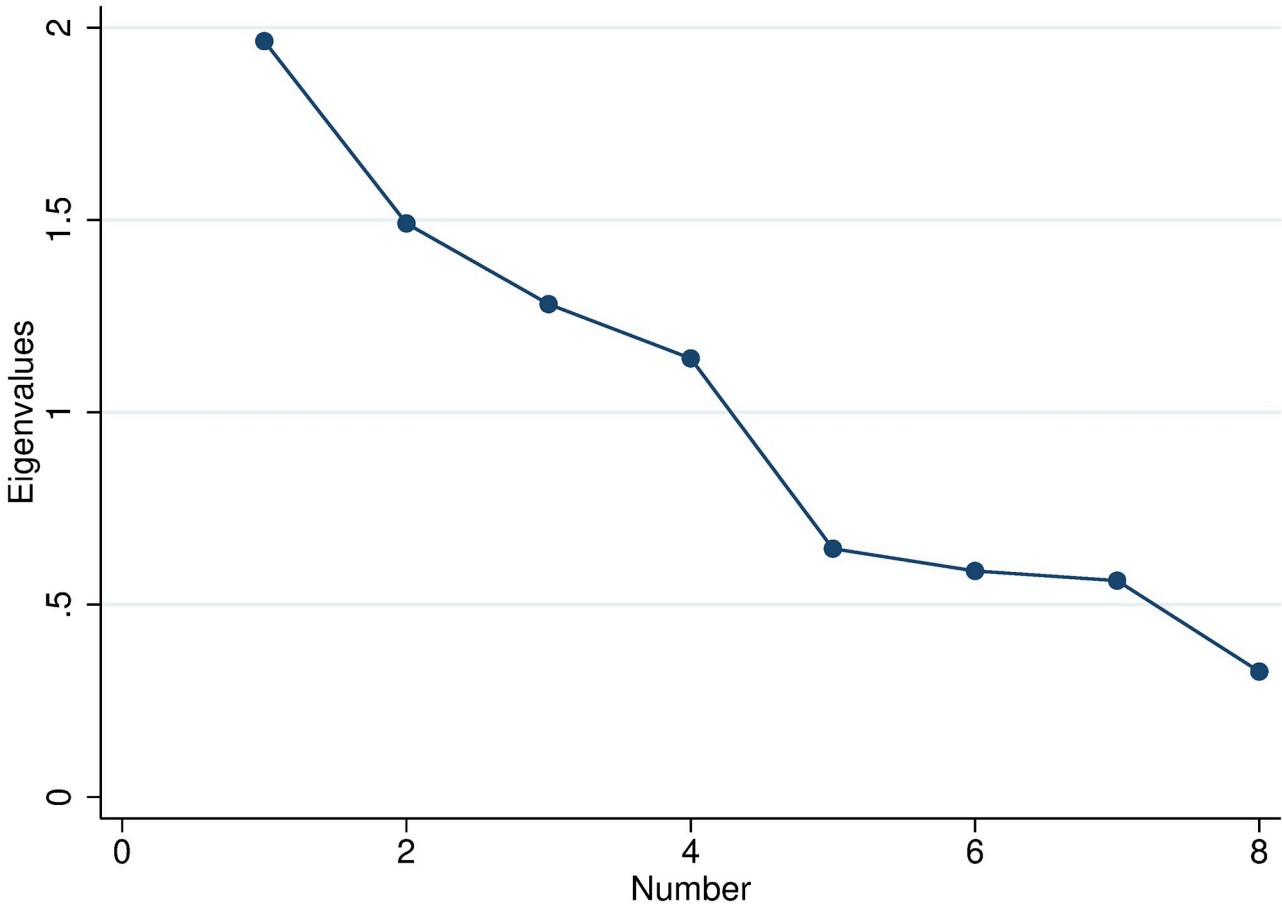

**Fig 5. Scree plot with eigenvalues.**

e.g. [54, 55]). Referring to Eq (1), **x** is a vector containing the estimations from our factor model and additional independent variables considered to be relevant to explain the classification outcome. We estimate a reduced (Table 5) and a full specification (Table 6), the latter containing the additional variables left out from the factor model. All estimations include time dummies, inventor countries and technology weights. We refrain from reporting tests on the

**Table 4. Rotated factor loadings (pattern matrix) and unique variances.**

| Variable | Factor 1 | Factor 2 | Factor 3 | Factor 4 | Uniqueness |
|---|---|---|---|---|---|
| lnclaims_ind | 0.3401 | 0.0510 | -0.0398 | **-0.7662** | 0.2931 |
| claims_per_ind_claim | 0.1677 | 0.0080 | -0.0084 | **0.8633** | 0.2264 |
| lnpatent_scope | 0.1889 | 0.0407 | **0.8631** | 0.0394 | 0.2161 |
| lnbwd_cits | **0.7635** | 0.2328 | -0.0800 | -0.0585 | 0.3530 |
| lnnpl_cits | **0.8066** | -0.0108 | 0.0608 | 0.0039 | 0.3456 |
| lntech_age | 0.2744 | -0.0072 | **-0.7896** | 0.0249 | 0.3005 |
| originality | 0.2900 | **0.8011** | 0.2360 | -0.0071 | 0.2185 |
| radicalness | -0.0571 | **0.9030** | -0.1093 | -0.0202 | 0.1689 |

Absolute loadings > 0.35 in bold

**Table 5. Multinomial logit reduced model—dependent variable: Patent category, base category: product patent.**

| | categorization based on all claims | | categorization based on only independent claims | |
|---|---|---|---|---|
| | process patent (1) | mixed patent (2) | process patent (3) | mixed patent (4) |
| Citations | 0.0877 *** | 0.282 *** | 0.0355 *** | 0.231 *** |
| | (0.00212) | (0.00149) | (0.00196) | (0.00155) |
| Originality and Radicalness | 0.113 *** | 0.101 *** | 0.108 *** | 0.110 *** |
| | (0.00172) | (0.00120) | (0.00161) | (0.00131) |
| Age and Scope | 0.109 *** | 0.0939 *** | 0.0985 *** | 0.0632 *** |
| | (0.00175) | (0.00130) | (0.00162) | (0.00141) |
| Claims | 0.0332 *** | -0.660 *** | 0.101 *** | -1.153 *** |
| | (0.00156) | (0.00160) | (0.00154) | (0.00183) |
| USPTO | 0.0647 *** | -0.859 *** | 0.00450 | -0.713 *** |
| | (0.00462) | (0.00329) | (0.00406) | (0.00372) |
| Constant | -0.677 *** | 0.628 *** | -0.572 ** | 0.147 |
| | (0.195) | (0.160) | (0.178) | (0.168) |
| Observations | 4800465 | | 4800110 | |
| Pseudo $R^2$ | 0.186 | | 0.204 | |
| chi2 | 1259919.8 | | 1343087.3 | |

Robust standard errors in parentheses.

* $p < 0.05$,

** $p < 0.01$,

*** $p < 0.001$. Time dummies, inventor country and technology shares included.

independence of irrelevant alternatives because those tests are susceptible to size bias and model complexity, which applies to our estimates because they are based on millions of observations (patents) and a large number of independent variables [56].

## Results

Tables 5 and 6 show coefficients for the specification with only factor scores and with other important characteristics, respectively. The results presented in here use product patents as base category. Estimates based on process and mixed patents as base categories can be found in the (S1 Table).

If a patent applicant or inventor were to add more citations of prior art, to invent more original or radical patents, or expand the technological scope in younger technologies, the multinomial log-odds for process and mixed patents relative to product patents would be expected to increase while holding all other variables in the model constant. A higher number of backward citations (Citations), higher originality and radicalness, and technological scope, as well as a younger technology (Age and Scope) thus increase the likelihood of getting either a process or mixed patent granted compared to the likelihood of getting a product patent. The association between Citations is particularly strong with mixed patents. This indicates that a higher reliance on predecessor inventions and research mainly leads to more mixed patents rather than product patents. The strong association between originality and radicalness with both process and mixed patents suggests that they are indeed more complex than pure product patents, because they not only refer to a broader range of IPC codes in their backward citations, but also deviate from prior technological paradigms. The result for technological scope points in the same direction. As the factor loading for technological age is negative, the estimation results suggest that the younger the technology, the more likely the granted patent is

**Table 6. Multinomial logit full model—dependent variable: Patent category, base category: product patent.**

| | categorization based on all claims | | categorization based on only independent claims | |
|---|---|---|---|---|
| | process patent (1) | mixed patent (2) | process patent (3) | mixed patent (4) |
| Citations | 0.0691 *** | 0.275 *** | 0.0228 *** | 0.226 *** |
| | (0.00217) | (0.00152) | (0.00201) | (0.00160) |
| Originality and Radicalness | 0.112 *** | 0.0923 *** | 0.109 *** | 0.102 *** |
| | (0.00172) | (0.00121) | (0.00161) | (0.00132) |
| Age and Scope | 0.0955 *** | 0.0874 *** | 0.0889 *** | 0.0544 *** |
| | (0.00183) | (0.00135) | (0.00169) | (0.00146) |
| Claims | 0.0283 *** | -0.652 *** | 0.0973 *** | -1.145 *** |
| | (0.00156) | (0.00161) | (0.00155) | (0.00184) |
| USPTO | 0.0835 *** | -0.851 *** | 0.0122 ** | -0.696 *** |
| | (0.00478) | (0.00343) | (0.00422) | (0.00388) |
| upward_tech_cycle | -0.0648 *** | -0.00653 *** | -0.0561 *** | 0.000445 |
| | (0.00236) | (0.00180) | (0.00218) | (0.00197) |
| lnfamily_size | 0.00263 | -0.0193 *** | -0.00493 * | 0.00813 *** |
| | (0.00236) | (0.00173) | (0.00220) | (0.00185) |
| lnnb_inventors | 0.251 *** | 0.200 *** | 0.182 *** | 0.0915 *** |
| | (0.00392) | (0.00288) | (0.00366) | (0.00311) |
| lnnumb_words_ind | 0.0513 *** | -0.470 *** | 0.135 *** | -0.545 *** |
| | (0.00438) | (0.00354) | (0.00402) | (0.00371) |
| Constant | -1.552 *** | 1.712 *** | -1.515 *** | 1.703 *** |
| | (0.196) | (0.161) | (0.179) | (0.169) |
| Observations | 4800271 | | 4799916 | |
| Pseudo $R^2$ | 0.189 | | 0.208 | |
| chi2 | 1272444.8 | | 1350628.1 | |

Robust standard errors in parentheses.

* $p < 0.05$,

** $p < 0.01$,

*** $p < 0.001$.

Time dummies, inventor country and technology shares included.

mixed or a process one. Thus, process and mixed patents are not only younger than product inventions, they also have a higher degree of novelty, which is most likely associated with higher risk in their development. Given the descriptive results above showing that the proportion of mixed patents has increased, these estimation results suggest that invention risk has also increased over time, at least in some technologies.

Claims is the only factor that shows a negative relationship with mixed patents, but a positive with process patents. The more dependent claims per independent claim a patent has, the more likely it will be a pure process patent, and the less likely a mixed patent compared to being a product patent. Conversely, the more independent claims a patent has, the less likely it will be a pure process patent, and the more likely a mixed patent. Even though the results must be interpreted with caution, a higher number of dependent claims per independent claims might reflect either high complexity, requirements from the examiners or strategic considerations. Van Zeebroeck, de la Potterie, & Guellec (2009) studied the contribution of the diffusion of national practices, technological complexity, emerging sectors and patenting strategies in explaining the number of claims of EPO patents [57]. Even though all elements are

important, they found that institutional influences and the international harmonization are the most important factors. Finally, filing a patent application at the USPTO reduces the likelihood of getting a mixed patent.

In an extended specification, we include further important variables to characterize the different patent categories. Filing a patent application in the upward cycle of a technology reduces the likelihood of getting a process patent granted. This seems to be in line with the theoretical framework that firms draw on processes at a later stage of the product cycle in order to sell into mass markets. But it is at odds with the result that younger technologies are associated with more process inventions. However, it could also indicate that process patents are more frequently found in technologies with shorter product life cycles, where technological maturity occurs earlier. This means that the product development opportunities are exhausted very soon and process inventions gain in importance. Furthermore, it could well be that patenting in younger technologies only in combination with technological complexity explains the strong correlation with process patenting, as indicated by the age and scope factor.

The size of the patent family is often associated with the commercialization potential of the invention. The positive correlation between mixed patents and family size in the case of independent claims and the negative correlation when all claims are classified may therefore indicate that mixed patents only have larger commercialization potential if both product and process characteristics are covered in the independent claims. In turn, adding process elements in the dependent claims seems to be associated with less commercialisation potential.

The size of the inventor team makes both process and mixed patents more likely. This might also reflect the findings that complex technologies such as software developments are typically based on larger teams as knowledge about those technologies can be transferred rather easily (for example, code can be shared on repositories). Finally, a higher number of words per independent claim decreases the likelihood of process patents and increases the likelihood of mixed patents. To recall, a higher number of words indicates less patent protection breadth. Therefore, pure process patents might be more difficult to protect than product inventions, which comes at no surprise and is mentioned in most survey articles on product and process innovations [15]. The result for mixed patent suggests that they can be protected easier than product patents which is much more surprising.

## Conclusions

The goal of this paper is to introduce a comprehensive classification of product and process patents based on keywords in claims and to provide descriptive evidence on potential relationships with fundamental patent characteristics. We show that the share of patents that contain both product and process claims (mixed patents) has increased tremendously. The share of pure process patents is still rather low. The trend towards including more and more process claims in addition to product claims can have many reasons, such as a larger complexity of the underlying technologies, a general technological exhaustion, strategic patenting behavior of large firms, or specific requirements from the patent offices and examiners. Generally, we observe large differences across technologies with computer technology having the largest share of mixed patents. As the patent premium also fluctuates widely across industries [58], such fluctuations might be also related to the industry's fraction of mixed and particularly process patents.

The resulting database can be of great use in order to study different aspects of the R&D and patenting process in more detail. For example, it enables researchers to distinguish between different types of knowledge accumulation in the framework of a knowledge production function, and to conduct more in-depth industry studies.

The characterization of product, process and mixed patents is an important contribution to different strands of empirical and theoretical literature and can contribute to a better understanding of R&D processes in companies. In particular, our analysis of relationships between fundamental patent characteristics and the different patent categories shows that younger technologies and potentially more complex technology combinations are associated with more process and mixed patents rather than product patents. Especially mixed patents seem to draw on already available knowledge and might be thus more science-based, original and radical than pure product patents. We do not find evidence that mixed patents are filed for mainly strategic reasons. A higher number of dependent claims per independent claim which may indicate strategic considerations or legal requirements seems to be associated with more process patents, but with less mixed patents. Our analysis confirms the notion that process inventions are more difficult to protect than product inventions, which is the reason why for many processes secrecy or lead-time advantages are the preferred appropriation mechanism.

A further finding is that larger inventor teams are more likely to engage in process-related research projects. Since the size of inventor teams is positively correlated with the likelihood of filing process patents rather than product or mixed patents, the positive relationship between inventor teams and performance which other studies found could be related to the fact that teams are more likely to develop process inventions. Thus, this study points to a previously neglected factor that could improve our understanding of the role of inventor teams in productivity. A plausible reason why larger inventor teams are more likely to engage in process-related research projects might be that processes are at the same time more complex and riskier than product inventions. Consequently, they might also have a greater market potential if the research project is successful which then translates into higher performance.

The study is subject to some limitations. First, in general, patents are not the preferred mechanism to appropriate newly created knowledge, especially for smaller firms [59, 60]). This is especially true for process inventions and implies that most of them remain unobservable. Second, although we attained robust classification results which were confirmed by robustness tests with different keyword combinations, we cannot completely rule out the possibility that additional keywords could slightly alter the results. Third, with only three patent types (product, process and mixed) that are used in our analysis we admittedly loose a lot of valuable information. Further research might try to uncover more nuances with respect to patents having different shares of process and product claims.

Further research might also develop and apply more advanced text mining and scraping methods and perhaps extend the scope beyond patent data. For example, product data on firms' websites or protocols from R&D labs might deliver more accurate pictures of the amount of process inventions. Finally, it is of particular interest to study diminishing returns to R&D given that it is getting harder to invent commercially successful technologies [61], a phenomenon that has caught attention in recent years. Our patent classification might provide useful metrics in order to study it.

## Supporting information

**S1 File. Appendix.** This appendix describes the implementation of the keyword search, the identification of independent and product-by-process claims, and the construction of a measure for growing technologies.
(PDF)

**S1 Table. Regression results with different basis.** These tables contain the results from the multinomial logit model with different base categories.
(PDF)

## Author Contributions

**Conceptualization:** Sebastian Heinrich, Florian Seliger, Martin Wörter.

**Data curation:** Florian Seliger.

**Formal analysis:** Sebastian Heinrich, Florian Seliger.

**Funding acquisition:** Florian Seliger, Martin Wörter.

**Investigation:** Sebastian Heinrich, Florian Seliger.

**Methodology:** Sebastian Heinrich, Florian Seliger.

**Project administration:** Florian Seliger.

**Supervision:** Florian Seliger.

**Visualization:** Sebastian Heinrich, Florian Seliger.

**Writing – original draft:** Sebastian Heinrich, Florian Seliger.

**Writing – review & editing:** Sebastian Heinrich, Florian Seliger.

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
