## [Decision Letter · Decision Letter 0]

4 Aug 2021

PONE-D-21-18643

Identifying and characterising product and process inventions in patent data

PLOS ONE

Dear Dr. Seliger,

Thank you for submitting your manuscript to PLOS ONE. After careful consideration, we feel that it has merit but does not fully meet PLOS ONE’s publication criteria as it currently stands. Therefore, we invite you to submit a revised version of the manuscript that addresses the points raised during the review process.

The review process for your manuscript entitled "Identifying and characterizing product and process inventions in patent data," which you submitted to PLOS ONE, is now complete. I have received comments from two reviewers. The review team thoroughly reviewed the manuscript and both reviewers recommend major revision. Please read carefully the comments and submit the revised manuscript.

We look forward to receiving your revised manuscript.

Kind regards,

Wonjoon Kim, Ph.D

Academic Editor

PLOS ONE

Journal Requirements:

We are grateful to the European Academic Research Programme for the financial support of this project.

We received funding from the European Patent Office Academic Research Programme: https://www.epo.org/learning/materials/academic-research-programme.html

Part of Sebastian Heinrich's salary was funded by this grant.

NO - The funders had no role in study design, data collection and analysis, decision to publish, or preparation of the manuscript.

Additionally, because some of your funding information pertains to [commercial funding//patents], we ask you to provide an updated Competing Interests statement, declaring all sources of commercial funding. 

In your Competing Interests statement, please confirm that your commercial funding does not alter your adherence to PLOS ONE Editorial policies and criteria by including the following statement: "This does not alter our adherence to PLOS ONE policies on sharing data and materials.” as detailed online in our guide for authors  http://journals.plos.org/plosone/s/competing-interests.  If this statement is not true and your adherence to PLOS policies on sharing data and materials is altered, please explain how. 

Please include the updated Competing Interests Statement and Funding Statement in your cover letter. We will change the online submission form on your behalf.

4. We noted in your submission details that a portion of your manuscript may have been presented or published elsewhere. [http://documents.epo.org/projects/babylon/eponet.nsf/0/A69432F980D71284C12584F5003DE05C/$File/ARP_report_Woerter_en.pdf] Please clarify whether this publication was peer-reviewed and formally published. If this work was previously peer-reviewed and published, in the cover letter please provide the reason that this work does not constitute dual publication and should be included in the current manuscript.

Reviewers' comments:

Reviewer's Responses to Questions

**Comments to the Author**

1. Is the manuscript technically sound, and do the data support the conclusions?

Reviewer #1: No

Reviewer #2: Yes

2. Has the statistical analysis been performed appropriately and rigorously? 

Reviewer #1: Yes

Reviewer #2: Yes

3. Have the authors made all data underlying the findings in their manuscript fully available?

Reviewer #1: Yes

Reviewer #2: Yes

4. Is the manuscript presented in an intelligible fashion and written in standard English?

Reviewer #1: Yes

Reviewer #2: Yes

5. Review Comments to the Author

Reviewer #1: The paper uses accessible patent data to classify patents into process patents, product patents, and mixed patents, which is an important study to promote the understanding of process patents, which are of particular interest in patent analysis. I will be happy to recommend the publication of the paper after the authors address a few points, some of which are quite important in my opinion.

My main concern is that the statistical validity of the method used to identify process patents, product patents and mixed patents, which is one of the core contributions of the study, is not ensured. The authors say that "In contrast, the identification of processes is quite complete if using the extracted keywords". However, I am not sure why only text information can be used to classify process patents when other information is available such as article citations and IPC. Moreover, the study uses keywords based on the patent's expert as queries in order to classify each patent, but there is no statistical explanation as to whether these keywords are appropriate enough to obtain a comprehensive list of process patents. The paper only obtains patents where the set keywords appear at the beginning of the sentence, and there must be sufficient evidence to show that these are equal to process patents, product patents or mixed patents. It would be good to compare several methods to show that the proposed method is a better one.

Also, the analysis of the word window used to classify each claim described in Supporting Information 1 needs to be tested to ensure the validity of the method, e.g. by checking how many words are identified as process patents (e.g. 1-10 words) rather than visually comparing 2-words and 5-words criterion.

With these experiments, it can be said that the characteristics of each patent have been clarified, as mentioned in the Conclusion.

And why not do the same analysis for process patents and mixed patents? Although this study compares process patents and mixed patents using product patents as a base, the comparison of the two relatively new forms of patents(mixed and process) will enable a more detailed description of the characteristics of each of the three patents.

Minor

In some figures, authors could make it more legible. For example, in Fig.1, what does the name of y-axis "Count in 1000" mean? Comparing the EPO and the USPTO, the USPTO has a larger scale on the y-axis, but where does this difference come from? To illustrate the percentage of each patent as mentioned in page 5, it would be easier to read if stacked bar graphs showing the percentage of each patent was added behind the plot.

In page 8 line 246, the word "nnlp_cits" must be "lnnpl_cits".

In Table1. process keywords and use keywords should be written separately.

Reviewer #2: This study focuses on identifying the product and process inventions and characterizing them using patent data. The results provide some meaningful technological trend and practical implications. However, there are several points needed to be addressed for more complete one.

1 This study lacks an academic contribution. The theme is quite technical so that the research motivation is more or less unclear. It means this study has a weak connection to theoretical literature. Simply identifying product and process patents and linking them to the fundamental characters of patents are not enough to publish in the academic journal. In that sense why this study is necessary has to be firmly addressed in the theoretical context: economic theory, technology management, innovation theory, and so on. Also, the authors need to shed light on the novelty of this study and its results compared to the previous studies and the relevant literature.

2 Similarly, in order to overcome the limited value of this study, interpreting the results should be extended to the theoretical literature. How they are meaningful in the technology management, innovation, industry trend, and so on should be addressed.

3 The conclusion part is too short to enrich its academic value. The present conclusion seems a repetition of the findings rather than a discussion on the main findings, their limitations, and further studies. Inclusion of discussion part is strongly recommended.

4. The present title seems to be quite neutral. Title modification is also recommended, which reflects its academic contribution.

5 There is no numbering of subtitles. It may lead the readers to a little confusion to follow the paper. It should be more systematic.

6. PLOS authors have the option to publish the peer review history of their article (what does this mean?). If published, this will include your full peer review and any attached files.

Reviewer #1: No

Reviewer #2: No

---

## [Author Response · Author response to Decision Letter 0]

26 Jan 2022

Editors’ comments:

Answer: We have adjusted the manuscript and naming of files according to the style templates.

Answer: There was no specific grant number for this funding program. If you need further information, let us kindly know it.

We are grateful to the European Academic Research Programme for the financial support of this project.

Please remove any funding-related text from the manuscript …

Answer: We have removed the Acknowledgements.

…and let us know how you would like to update your Funding Statement. 

Currently, your Funding Statement reads as follows: 

We received funding from the European Patent Office Academic Research Programme: https://www.epo.org/learning/materials/academic-research-programme.html

Part of Sebastian Heinrich's salary was funded by this grant.

NO - The funders had no role in study design, data collection and analysis, decision to publish, or preparation of the manuscript.

Answer: Please update the Funding Statement as follows:

We are grateful to the European Academic Research Programme for the financial support of this project.

Part of Sebastian Heinrich's salary was funded by this grant.

Additionally, because some of your funding information pertains to [commercial funding//patents], we ask you to provide an updated Competing Interests statement, declaring all sources of commercial funding. 

In your Competing Interests statement, please confirm that your commercial funding does not alter your adherence to PLOS ONE Editorial policies and criteria by including the following statement: "This does not alter our adherence to PLOS ONE policies on sharing data and materials.” as detailed online in our guide for authors http://journals.plos.org/plosone/s/competing-interests. If this statement is not true and your adherence to PLOS policies on sharing data and materials is altered, please explain how. 

Please include the updated Competing Interests Statement and Funding Statement in your cover letter. We will change the online submission form on your behalf.

Answer: We’ve included the statement in the cover letter.

4. We noted in your submission details that a portion of your manuscript may have been presented or published elsewhere. [http://documents.epo.org/projects/babylon/eponet.nsf/0/A69432F980D71284C12584F5003DE05C/$File/ARP_report_Woerter_en.pdf] Please clarify whether this publication was peer-reviewed and formally published. If this work was previously peer-reviewed and published, in the cover letter please provide the reason that this work does not constitute dual publication and should be included in the current manuscript.

Answer: As mentioned in the text, the project was conducted for the EPO Academic Research Programme which also provided funding. The project report contains a lot of detailed information on which we also needed to draw in this publication. This publication has a different topic than the very comprehensive report as we characterize product and process patens with patent indicators etc. However, it is not possible to avoid some overlap when it comes to describing the classification process and the database, which has been cited accordingly. The project report is not a peer-reviewed paper. It is solely a project report that has been published on the EPO’s website.

Reviewer #1: 

The paper uses accessible patent data to classify patents into process patents, product patents, and mixed patents, which is an important study to promote the understanding of process patents, which are of particular interest in patent analysis. I will be happy to recommend the publication of the paper after the authors address a few points, some of which are quite important in my opinion.

Answer: Dear reviewer,

Thank you for your thorough reading of the manuscript and your suggestions to improve it.

My main concern is that the statistical validity of the method used to identify process patents, product patents and mixed patents, which is one of the core contributions of the study, is not ensured. The authors say that "In contrast, the identification of processes is quite complete if using the extracted keywords". However, I am not sure why only text information can be used to classify process patents when other information is available such as article citations and IPC. 

Answer: Concerning the usage of other information such as citations or IPC, it appears to us that citations or IPCs do not inherently contain information about the difference between products and processes. 

We were indeed considering to use the IPC in order to classify the patents in the beginning of the project. However, classification (IPC, CPC or whatever system is used) is a means to index the content and to make it more accessible for the examiner when searching for prior art, but it doesn’t provide consistent information to classify patents into product and process inventions. We have been pointed to this issue by patent experts (e.g. from the Fraunhofer Institute for Systems and Innovation Research ISI) and were advised against using IPC. 

Nevertheless, we checked a sample of patents and came to the same conclusion, namely that it is not possible to classify patents unambiguously based on the IPC. In the IPC, there are classes or subclasses whose names seem to indicate that they must solely contain either product or process patents, respectively. But this is usually not the case and you will both find product patents in IPC sections that seem to only include processes, and conversely. 

Claims, in turn, express the scope of protection sought (patent application) or actually obtained (granted patent). We find this kind of information to be more useful for our classification purposes, especially because the EPO and USPTO also have detailed examination guidelines where they define product and process claims in detail. We also want to point out that by using full-text data we can utilize much more and richer information than by relying on predefined metrics.

Finally, backward citations are the “prior art”, and mostly also have classification codes (patents for sure/NPL sometimes). We did not consider using citation data etc. for the same reasons. If we used citations, we would have needed to know if the cited patent is a process patent. But we do not know this ex ante, which is the reason why we propose a text-based classification for all patents.

Moreover, the study uses keywords based on the patent's expert as queries in order to classify each patent, but there is no statistical explanation as to whether these keywords are appropriate enough to obtain a comprehensive list of process patents. 

Answer: The keywords are appropriate with regard to content in the sense that it is possible to find almost all process patents according to the examination guidelines. In addition, research assistants classified the claims from 1100 patents manually and their results were pretty close to the results from keyword search. We also manually inspected about 1000 patent texts in the aftermath and came to the same conclusion.

The paper only obtains patents where the set keywords appear at the beginning of the sentence, and there must be sufficient evidence to show that these are equal to process patents, product patents or mixed patents. It would be good to compare several methods to show that the proposed method is a better one.

Answer: We actually compared the keyword-based classification with a classification from a machine learning approach. The results are largely similar in their classification outcome, but by extensive manual inspection, we found that the keyword approach captures the class boundaries better. We’ve added a paragraph briefly describing the machine learning approach and the reason for focusing on the keyword approach.

Also the analysis of the word window used to classify each claim described in Supporting Information 1 needs to be tested to ensure the validity of the method, e.g. by checking how many words are identified as process patents (e.g. 1-10 words) rather than visually comparing 2-words and 5-words criterion.

With these experiments, it can be said that the characteristics of each patent have been clarified, as mentioned in the Conclusion.

Answer: With the visual comparison, we wanted to show that it doesn’t make much difference whether we use the first 2 or first 5 words. The criteria were chosen after inspecting thousands of claim texts manually and also the resulting classification were checked against thousands of claim texts (see also explanations above).

From manual inspection, it has become clear that we need to limit the classification to the first 2 (minimum) or first 5 words (maximum) of a claim text (we could also have tried 3 or 4, but we stuck to the boundaries).Please also see the detailed examples in S1.

We computed the accuracy based on the training dataset that has been labeled for the machine learning approach (see above). The accuracy was very high for the keyword search based on either the first 2 or 5 words (98). Therefore, we are convinced that the patent claims were classified correctly.

And why not do the same analysis for process patents and mixed patents? Although this study compares process patents and mixed patents using product patents as a base, the comparison of the two relatively new forms of patents(mixed and process) will enable a more detailed description of the characteristics of each of the three patents.

Answer:Many thanks for this advice. We now present additional tables with process patents as the base category to compare mixed with process patents directly. The advantage of presenting the result in this way is that it is easier to see the significant differences between process and mixed patents. Note that we still discuss the results with respect to the base category product patents in order to keep the Results section readable. 

Minor 

In some figures, authors could make it more legible. For example, in Fig.1, what does the name of y-axis "Count in 1000" mean? Comparing the EPO and the USPTO, the USPTO has a larger scale on the y-axis, but where does this difference come from? To illustrate the percentage of each patent as mentioned in page 5, it would be easier to read if stacked bar graphs showing the percentage of each patent was added behind the plot.

Answer: Fig. 1 to 4 depict simple patent counts and the y-axis expresses counts of granted patents in 1000. We have adjusted the axes labels for more clarity. The difference in scale comes from the discrepancy in patent volume between the two offices. We decided not to rescale the EPO plots’ axes, as the time series would have been more compressed and therefore less able to deliver the required information.

Concerning stacked bar graphs, we think that time trends can be better depicted in line charts, especially the relative changes between the different categories. We think that the line plots make the comparison of differences in time trends more easily understandable, while we still plot the total patent counts. If this request is of high importance to you, we can of course provide the figures in the suggested format.

In page 8 line 246, the word "nnlp_cits" must be "lnnpl_cits".

In Table1. process keywords and use keywords should be written separately.

Answer: We have corrected the error and changed the table according to your suggestion. Thank you. 

Reviewer #2: 

This study focuses on identifying the product and process inventions and characterizing them using patent data. The results provide some meaningful technological trend and practical implications. However, there are several points needed to be addressed for more complete one.

1 This study lacks an academic contribution. The theme is quite technical so that the research motivation is more or less unclear. It means this study has a weak connection to theoretical literature. Simply identifying product and process patents and linking them to the fundamental characters of patents are not enough to publish in the academic journal. In that sense why this study is necessary has to be firmly addressed in the theoretical context: economic theory, technology management, innovation theory, and so on. Also, the authors need to shed light on the novelty of this study and its results compared to the previous studies and the relevant literature.

Answer: Thanks a lot for your careful reading of the paper and your suggestions to improve it. We absolutely agree that the embedding in the literature should be expanded so that the contribution of the study can be better assessed beyond the pure classification of the patents. We had paid too little attention to this. In the new version of the manuscript, we have significantly expanded this part. In the new manuscript, we show in the introduction in which contexts the classification of patents can make an important contribution to theoretical and empirical literature without distracting the reader from the main contribution of the study. 

2 Similarly, in order to overcome the limited value of this study, interpreting the results should be extended to the theoretical literature. How they are meaningful in the technology management, innovation, industry trend, and so on should be addressed.

Answer: We now discuss the results against the background of the existing literature and emphasize the contribution of this study in the Conclusions section. 

3 The conclusion part is too short to enrich its academic value. The present conclusion seems a repetition of the findings rather than a discussion on the main findings, their limitations, and further studies. Inclusion of discussion part is strongly recommended.

Answer: We have extended the conclusions according to your suggestion and, most importantly, added a discussion of the results. We also present limitations and suggestions for further studies.

Note that the Results section also discusses results to some extent.

4. The present title seems to be quite neutral. Title modification is also recommended, which reflects its academic contribution.

Answer: We’ve changed the title to ‘Appropriability and basicness of R&D – identifying and characterizing product and process inventions in patent data’, We hope that you share our view that it reflects the contribution better. 

5 There is no numbering of subtitles. It may lead the readers to a little confusion to follow the paper. It should be more systematic.

Answer: We have used the PlosOne Latex Template and followed all guidelines there. Articles in PlosOne don’t have a numbering of subtitles. Unfortunately, there is nothing we can do here.

---

## [Decision Letter · Decision Letter 1]

15 Jul 2022

Appropriability and basicness of R&D: Identifying and characterising product and process inventions in patent data

PONE-D-21-18643R1

Dear Dr. Seliger,

We’re pleased to inform you that your manuscript has been judged scientifically suitable for publication and will be formally accepted for publication once it meets all outstanding technical requirements.

Kind regards,

George Vousden

Staff Editor

PLOS ONE

Additional Editor Comments (optional):

Reviewers' comments:

Reviewer's Responses to Questions

**Comments to the Author**

1. If the authors have adequately addressed your comments raised in a previous round of review and you feel that this manuscript is now acceptable for publication, you may indicate that here to bypass the “Comments to the Author” section, enter your conflict of interest statement in the “Confidential to Editor” section, and submit your "Accept" recommendation.

Reviewer #2: (No Response)

2. Is the manuscript technically sound, and do the data support the conclusions?

Reviewer #2: Yes

3. Has the statistical analysis been performed appropriately and rigorously? 

Reviewer #2: Yes

4. Have the authors made all data underlying the findings in their manuscript fully available?

Reviewer #2: Yes

5. Is the manuscript presented in an intelligible fashion and written in standard English?

Reviewer #2: Yes

6. Review Comments to the Author

Reviewer #2: This revision is much improved than the previous one in providing its theoretical meaning. And also the title has been changed and matched well with it. Now it seems more proper for the academic journal. For more complete paper I’d like to suggest that the abstract should be changed by adding the theoretical contribution.

7. PLOS authors have the option to publish the peer review history of their article (what does this mean?). If published, this will include your full peer review and any attached files.

Reviewer #2: No

---

## [Editor Report · Acceptance letter]

5 Aug 2022

PONE-D-21-18643R1 

Appropriability and basicness of R&D: Identifying and characterising product and process inventions in patent data 

Dear Dr. Seliger:

I'm pleased to inform you that your manuscript has been deemed suitable for publication in PLOS ONE. Congratulations! Your manuscript is now with our production department. 

Kind regards, 

on behalf of

Dr. George Vousden 

Staff Editor

PLOS ONE